# The LuxO-OpaR quorum-sensing cascade differentially controls Vibriophage VP882 lysis-lysogeny decision making in liquid and on surfaces

Francis J. Santoriello[1], Bonnie L. Bassler[1,2]*

1 Department of Molecular Biology, Princeton University, Princeton, New Jersey, United States of America,
2 Howard Hughes Medical Institute, Chevy Chase, Maryland, United States of America

* bbassler@princeton.edu

**Data Availability Statement:** The authors confirm that all data underlying the findings are fully available without restriction. All relevant data are within the manuscript and Supporting Information

## Abstract

Quorum sensing (QS) is a process of cell-to-cell communication that bacteria use to synchronize collective behaviors. QS relies on the production, release, and group-wide detection of extracellular signaling molecules called autoinducers. Vibrios use two QS systems: the LuxO-OpaR circuit and the VqmA-VqmR circuit. Both QS circuits control group behaviors including biofilm formation and surface motility. The Vibrio parahaemolyticus temperate phage φVP882 encodes a VqmA homolog (called VqmAφ). When VqmAφ is produced by φVP882 lysogens, it binds to the host-produced autoinducer called DPO and launches the φVP882 lytic cascade. This activity times induction of lysis with high host cell density and presumably promotes maximal phage transmission to new cells. Here, we explore whether, in addition to induction from lysogeny, QS controls the initial establishment of lysogeny by φVP882 in naïve host cells. Using mutagenesis, phage infection assays, and phenotypic analyses, we show that φVP882 connects its initial lysis-lysogeny decision to both host cell density and whether the host resides in liquid or on a surface. Host cells in the low-cell-density QS state primarily undergo lysogenic conversion. The QS regulator LuxO~P promotes φVP882 lysogenic conversion of low-cell-density planktonic host cells. By contrast, the ScrABC surface-sensing system regulates lysogenic conversion of low-cell-density surface-associated host cells. ScrABC controls the abundance of the second messenger molecule cyclic diguanylate, which in turn, modulates motility. The scrABC operon is only expressed when its QS repressor, OpaR, is absent. Thus, at low cell density, QS-dependent derepression of scrABC drives lysogenic conversion in surface-associated host cells. These results demonstrate that φVP882 integrates cues from multiple sensory pathways into its lifestyle decision making upon infection of a new host cell.

## Author summary

Bacteria in nature often exist in surface-associated communities including sessile biofilms and highly motile swarms. Thus, bacteriophages can encounter their hosts in structured

files. Data points used to make plots have been provided as S1 Data.

**Funding:** The authors acknowledge financial support from the Howard Hughes Medical Institute, the National Institutes of Health grant R37GM065859, and the National Science Foundation grant MCB-2043238 (B.L.B.). The funders had no role in study design, data collection and analysis, decision to publish, or preparation of the manuscript.

**Competing interests:** The authors have declared that no competing interests exist.

communities. Much bacteriophage research is performed in homogenous, planktonic cultures containing cells that neither display the gene expression patterns nor the behaviors that occur in surface communities. The *Vibrio parahaemolyticus* temperate phage φVP882, after lysogenizing its host, can monitor the vicinal cell density and time lytic induction with high host cell density. Here, we show that, upon infection of a new host cell, φVP882 assesses host cell density to make the decision whether to lyse or lysogenize. Host cells at low density primarily undergo lysogenic conversion, and the components driving the phage decision-making process vary depending on whether the host cell is in liquid or associated with a solid surface. We propose that by tuning its lysis-lysogeny decision making to both host cell density and the physical environment of the host, φVP882 can maximize transmission to new host cells and dispersal to new environments.

## Introduction

Quorum-sensing (QS) communication between bacteria dictates whether cells undertake individual or group behaviors. QS relies on the production, release, accumulation, and detection of extracellular signaling molecules called autoinducers, and the process allows bacteria to synchronize collective activities [1,2]. The model QS bacterium and pathogen *Vibrio parahaemolyticus* uses two QS systems. The first system is the LuxO-OpaR cascade (Fig 1A), in which three autoinducers (called AI-1, AI-2, and CAI-1) are detected by membrane-bound receptors that modulate the phosphorylation state of LuxO to control production of non-coding RNAs called the Qrr sRNAs [3–8]. At low cell density, LuxO is phosphorylated (LuxO~P), the Qrr sRNAs are produced, and they repress production of the high-cell-density master regulator OpaR. Under this condition, *V. parahaemolyticus* cells act as individuals. At high cell density (the mode depicted in Fig 1A), autoinducer detection drives dephosphorylation of LuxO, the Qrr sRNAs are not made, and OpaR is produced. OpaR controls a regulon of genes underpinning group behaviors [9–11]. The second QS system consists of the cytoplasmic autoinducer receptor and transcription factor VqmA (Fig 1A), which binds the autoinducer called DPO and drives the production of a sRNA called VqmR. VqmR, in turn, controls genes required for group behaviors [12,13].

When *V. parahaemolyticus* associates with a solid surface, it can embark on one of two QS-controlled lifestyles: swarming motility at low cell density or biofilm formation at high cell density [9,10]. OpaR regulates these behaviors directly and indirectly, the latter through the ScrABC sensory pathway ([14–16]; and Fig 1B). The *scrABC* operon encodes the aminotransferase ScrA, periplasmic binding protein ScrB, and diguanylate cyclase/phosphodiesterase ScrC. ScrA produces a molecule known as the S-signal that binds ScrB [16,17]. In the absence of S-signal, ScrC functions as a diguanylate cyclase, increasing the intracellular pool of cyclic diguanylate (c-di-GMP) to induce exopolysaccharide production and biofilm formation [15,18,19]. ScrB bound to the S-signal interacts with ScrC on the inner membrane, which triggers the ScrC phosphodiesterase activity. Degradation of c-di-GMP reduces the c-di-GMP pool, which activates production of lateral flagella and swarming motility [15,16]. Other phosphodiesterases (ScrG and TpdA) and diguanylate cyclases (ScrJ, ScrL, and GefA) also modulate these behaviors, with ScrABC functioning as the primary controller of c-di-GMP abundance during growth on a surface [20–24]. Regarding the connection between QS and surface sensing, OpaR represses *scrABC* and thus, suppresses swarming motility at high cell density ([10]; and Fig 1B). This OpaR function drives an increase in the c-di-GMP pool and a shift toward biofilm formation. During surface growth, OpaR also represses the *lafK* gene encoding the

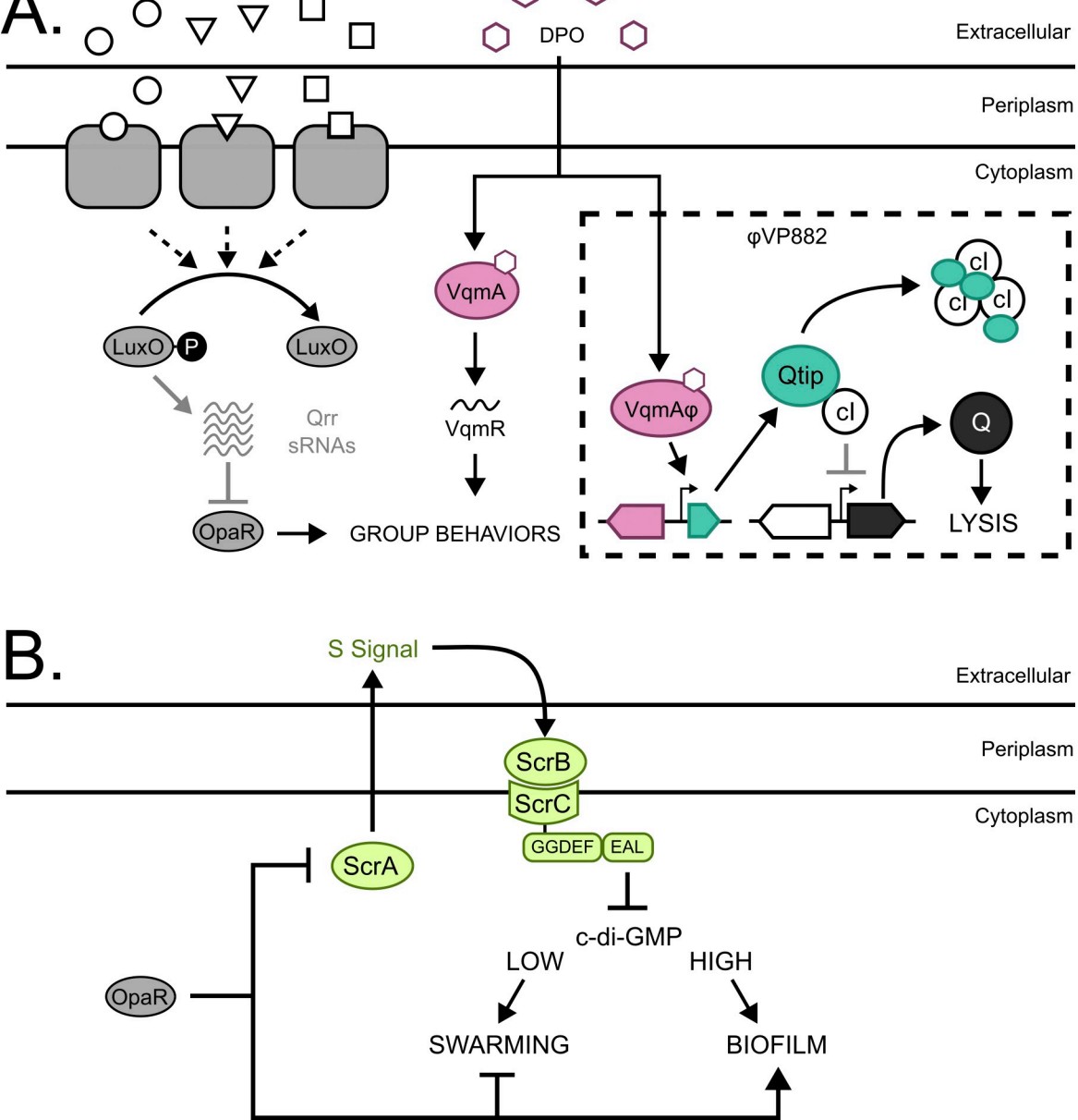

**Fig 1. *V. parahaemolyticus* QS controls φVP882 lysogeny-lysis transitions and surface sensing.** (A) Schematic of *V. parahaemolyticus* QS pathways and their known interactions with φVP882. Proteins involved in the LuxO-OpaR cascade are shown in gray. The circles, squares, and triangles represent the autoinducers AI-1, AI-2, and CAI-1 that are recognized by their corresponding receptors. At low cell density, autoinducers are sparse and LuxO is phosphorylated (LuxO~P). LuxO~P activates expression of the genes encoding the Qrr sRNAs, and the Qrr sRNAs repress *opaR*. At high cell density, LuxO is dephosphorylated and inactive. The Qrr sRNAs are not made, and OpaR is produced. The DPO-VqmA QS pathway is shown in magenta. DPO bound to the DPO autoinducer (depicted as hexagons) activates expression of the gene encoding the VqmR sRNA. OpaR and VqmR regulate genes underlying group behaviors. The dashed square highlights φVP882 and its key regulatory components. The φVP882 homolog of host VqmA, VqmAφ, also binds DPO, leading to production of Qtip, sequestration of the cI repressor, de-repression of the antiterminator Q, and activation of the φVP882 lytic cycle. (B) Schematic of the *V. parahaemolyticus* surface-sensing pathway. Upon surface association, ScrA produces an unknown molecule called the S-signal, which binds ScrB in the periplasm. ScrB in complex with S-signal associates with ScrC on the inner membrane, triggering ScrC phosphodiesterase activity which reduces the abundance of c-di-GMP and induces swarming motility. OpaR represses *scrABC* and, thus, swarming motility. By contrast, OpaR activates biofilm formation genes. GGDEF and EAL denote the ScrC diguanylate cyclase and phosphodiesterase enzymatic activities, respectively.

transcriptional activator of the lateral flagellar (*laf*) operon while OpaR activates the *cpsR* and *cpsQ* genes encoding transcriptional activators of the exopolysaccharide (*cps*) operon [11,18,25–27]. The net effect of these OpaR functions is suppression of swarming and activation of biofilm formation at high cell density.

Phages in nature encounter and infect bacterial hosts that exist as individual planktonic cells or in surface-attached biofilm communities [28,29]. Following infection of a host cell, a temperate phage either replicates and lyses the host or enters a dormant state of genome maintenance called lysogeny [30–32]. It is known that in model temperate phages such as phage λ, the outcome of the lysis-lysogeny decision depends on the multiplicity of infection (MOI). Specifically, the higher the MOI, the more lysogeny that occurs [33,34]. In the lysogenic state, the phage genome, or prophage, either integrates into the host genome (e.g., λ) or is maintained as an extrachromosomal plasmid-like element (e.g., P1, N15) [32,35,36]. Lysogens can activate and become lytic. Traditionally, lytic induction was understood to occur exclusively in response to host cell stress such as DNA damage [31,32,37,38]. New research has upended this dogma by revealing that phages can monitor host sensory cues and exploit the information they garner to drive lysogeny-lysis transitions [39–43]. The *V. parahaemolyticus* plasmid-like phage VP882 (called φVP882) "eavesdrops" on host QS communication via a phage-encoded homolog of VqmA (called VqmAφ) that allows the phage to monitor the abundance of host DPO ([39]; and Fig 1A). At high host cell density, VqmAφ binds to DPO and activates production of the phage anti-repressor Qtip that sequesters the phage cI repressor and triggers production of the lytic regulator Q, and Q promotes host cell lysis. Presumably, lysing the host exclusively at high cell density allows the phage to maximize transmission to new hosts. VqmAφ-dependent induction of φVP882 from lysogeny is also regulated by the LuxO-OpaR QS pathway [43]. Specifically, host *V. parahaemolyticus* strains with functional QS systems express higher levels of the φVP882 lytic genes than strains locked in the low-cell-density QS state. These results indicate that multiple QS pathways can influence φVP882 lifestyle transitions.

While QS clearly drives φVP882 lytic induction, the initial φVP882 commitment to lysis or lysogeny that occurs upon infection of a naïve host has not been studied. Here, we investigated whether the *V. parahaemolyticus* LuxO-OpaR QS system plays a role in this initial φVP882 lysis-lysogeny decision. Further, given the known connection between OpaR, surface sensing, and surface group behaviors including swarming and biofilm formation, we explored whether the host's physical environment influences the initial φVP882 lysis-lysogeny decision. Specifically, we quantified lysogeny and phage virion production during φVP882 infection of wild-type (WT) *V. parahaemolyticus* and a panel of *V. parahaemolyticus* QS mutants in both planktonic and surface-associated contexts. Our results demonstrate that the initial φVP882 lysis-lysogeny decision is regulated by the host LuxO-OpaR QS system. First, the phage shows an extreme preference for establishing lysogeny in cells that exist in the low-cell-density QS state, both in liquid and on a surface. Regarding low-cell-density planktonic host cells, lysogeny is driven by LuxO~P through the Qrr sRNAs via an OpaR-independent mechanism. In low-cell-density surface-associated host cells, however, lysogeny is driven by the absence of OpaR, which enables derepression of the surface-sensing operon–*scrABC*. ScrABC production leads to a decrease in cytoplasmic c-di-GMP which, in turn, promotes increased φVP882-driven lysogenic conversion. Importantly, these patterns are not due to an inability of φVP882 to infect *V. parahaemolyticus* cells at high cell density, as infection of such cells leads to robust phage particle production. Our findings demonstrate that φVP882 can incorporate cues from three different *V. parahaemolyticus* sensory pathways (VqmA-VqmR, LuxO-OpaR, and ScrABC), allowing it to align its lifestyle transitions with host cell density and particular environmental conditions.

## Results

### *V. parahaemolyticus* QS mutants for investigation of φVP882 infection outcomes

*V. parahaemolyticus* O3:K6 strain RIMD2210633 (from here forward called RIMD) encodes functional versions of all known QS components. To assess what effects host QS and down-stream changes in c-di-GMP abundance have on φVP882 infection outcomes, we constructed mutants that lock RIMD into the low-cell-density and high-cell-density QS states. The low-cell-density-locked strains are $luxO^{D61E}$, $\Delta opaR$, and the double $luxO^{D61E} \Delta opaR$ and $luxO^{D61A}$ $\Delta opaR$ mutants. The high-cell-density-locked strain is $luxO^{D61A}$. The logic is as follows (see Fig 1A): LuxO~P exists at low cell density and dephosphorylated LuxO exists at high cell density. LuxO$^{D61E}$ is a LuxO phosphomimetic and LuxO$^{D61A}$ cannot be phosphorylated. Thus, the $luxO^{D61E}$ strain is locked into the low-cell-density QS mode because it constitutively produces the Qrr sRNAs, and the $luxO^{D61A}$ strain is locked into the high-cell-density QS mode because it lacks Qrr sRNA production. OpaR drives the high-cell-density QS program, and thus, the $\Delta opaR$ strain is locked into the low-cell-density state. Likewise, because OpaR functions down-stream of LuxO and the Qrr sRNAs, the $luxO^{D61E} \Delta opaR$ and $luxO^{D61A} \Delta opaR$ double mutant strains are also locked into the low-cell-density QS mode. These two strains differ in that the $luxO^{D61E} \Delta opaR$ strain produces the Qrr sRNAs and the $luxO^{D61A} \Delta opaR$ strain does not. This difference will become important below. Each of the above mutants displayed the expected low-cell-density or high-cell-density pattern of expression of the known OpaR-activated gene *luxC* ($P_{luxC}$) (S1A Fig). Moreover, the mutants demonstrated the expected biofilm and swarm-ing motility phenotypes–swarming when locked in low-cell-density mode and biofilm forma-tion when locked in high-cell-density mode (S1B and S1C Fig, respectively). All mutations were constructed in RIMD lacking exopolysaccharide production ($\Delta cpsA$) and polar flagellar motility ($\Delta pomA$) as both are known to inhibit infection by some phages [44,45]. Eliminating these components enabled us to investigate φVP882 infection and its lysis-lysogeny transitions without interference from obvious physical impediments to infection.

### An assay to quantify post-infection levels of φVP882 lysis and lysogeny of RIMD strains

To investigate the role of QS in the initial φVP882 lysis-lysogeny decision upon infection of a naïve RIMD host, we developed assays to quantify post-infection levels of lytic replication and lysogenic conversion. φVP882 lytic replication was determined using quantitative polymerase chain reactions (qPCR) against the φVP882 gene *gp69* on DNase-treated, cell-free culture flu-ids collected from infected populations. This method enabled quantitation of capsid bound viral genomes at the beginning and end of the φVP882 infection period. We used the values to determine the amount of lytic replication that occurred in a population. Regarding lysogenic conversion, for reasons we do not understand, insertion of an antibiotic resistance cassette into the φVP882 genome resulted in a strong defect in phage infectivity. Thus, simple enumer-ation of antibiotic resistant colonies to quantify lysogenic conversion post infection was not possible. To circumvent this issue, we leveraged the φVP882 major lytic regulator Q (Fig 1A). RIMD strains carrying an arabinose-inducible copy of *q* on a plasmid ($P_{bad}$-*q*) were infected with φVP882 in medium supplemented with $CaCl_2$, a requirement for phage adsorption and subsequent viral entry. After 24 h, the cells were collected, diluted into fresh medium lacking $CaCl_2$, and Q production was induced. Cells that had been lysogenized by φVP882 lysed while uninfected cells did not die (Fig 2A). Removal of $CaCl_2$ prior to induction of *q* ensured no sub-sequent rounds of infection occurred. Thus, counterintuitively, quantitation of lysis could be

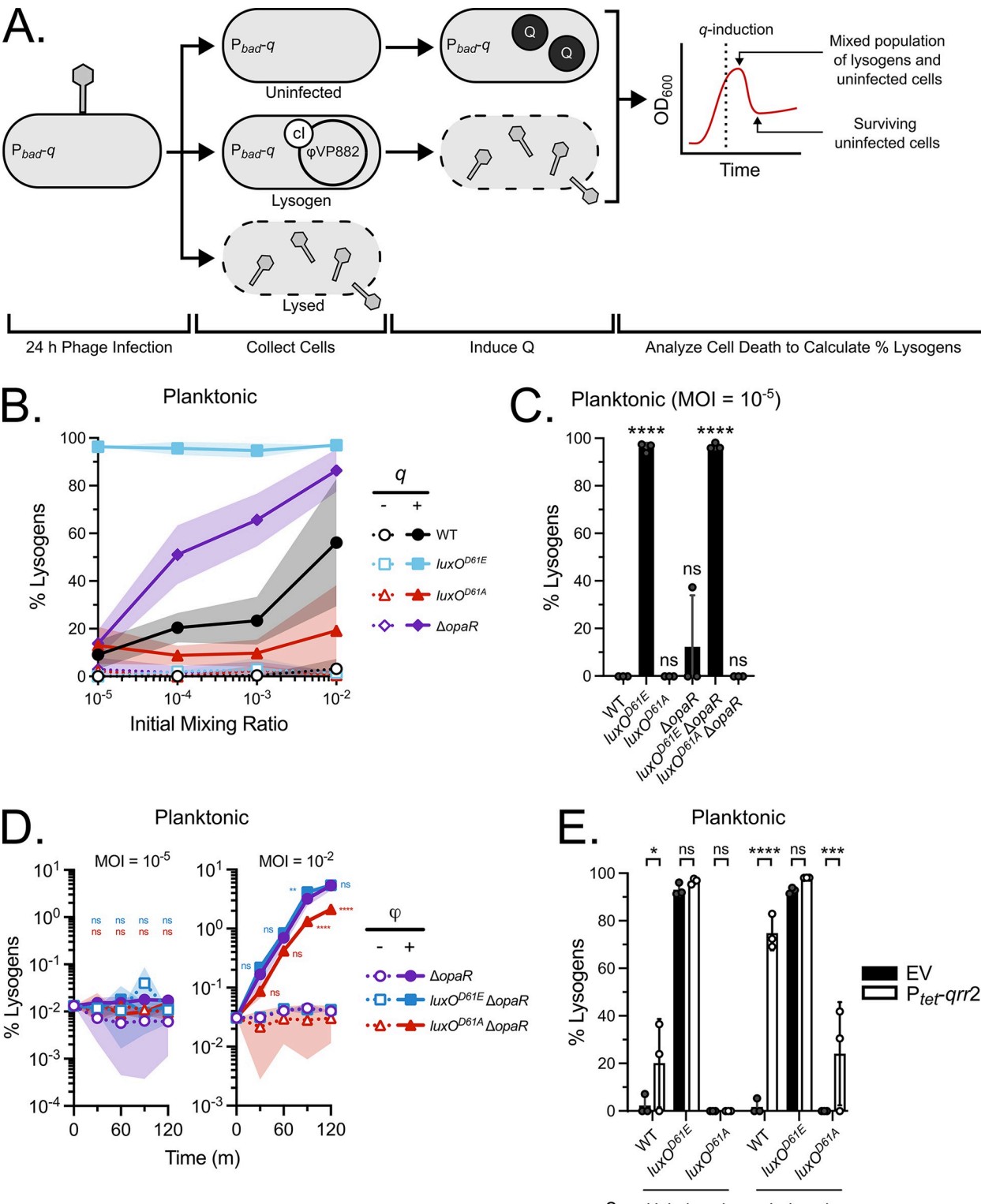

**Fig 2. The Qrr sRNAs control φVP882 lysogenic conversion of planktonic *V. parahaemolyticus* RIMD strains.** (A) Schematic of lysogenic conversion quantitation by *q*-induction. Following addition of φVP882 for 24 h to RIMD carrying P*bad*-*q*, three outcomes are possible: Cells can remain uninfected or naïve (top), become infected and lysogenized (middle), or become infected and lyse (bottom). Upon induction of *q* in the cells that survived infection, naïve cells remain unaffected while lysogenized cells lyse. Thus, cell death increases with the proportion of lysogens in each culture. The outcome of this assay is a peak and a subsequent valley in optical density of the culture (depicted in the theoretical plot on the right).

The difference between the optical density values at the peak (total cells) and valley (remaining uninfected cells) can be used to determine the percentage of the population that has been lysogenized (see Materials and Methods). (B) Percent lysogens in the indicated planktonic strains at different MOIs. (C) Percent lysogens in the indicated planktonic strains. Infections were performed at MOI = $10^{-5}$. (D) Lysogenic conversion over time in the indicated strains at (left) MOI = $10^{-5}$ and (right) MOI = $10^{-2}$. Significance notations indicate comparisons between the $luxO^{D61E}$ $\Delta opaR$ strain (blue) or the $luxO^{D61A}$ $\Delta opaR$ strain (red) and the $\Delta opaR$ strain. (E) Percent lysogens in the indicated planktonic strains following overexpression of $qrr2$ (EV = empty vector, P$_{tet}$-$qrr2$ = inducible $qrr2$) during infection. Expression of $qrr2$ was uninduced (-aTc) or induced (+aTc). Infections were performed at MOI = $10^{-5}$. (B-E) All experiments were performed in biological triplicate (n = 3). (B,D) Lines and symbols represent the means and shaded areas represent the standard deviations. (C,E) Symbols represent individual replicate values. Bars represent the means. Error bars represent standard deviations. Results with $q$-induction (+arabinose) are shown. (C,D,E) Significance was determined by two-way ANOVA with Tukey's multiple comparisons test to determine adjusted p-values: (C) ns = non-significant, **** p < 0.0001, (D) ns = non-significant, ** p = 0.0049, **** p < 0.0001, (E) ns = non-significant, * p = 0.0158, *** p = 0.0004, **** p < 0.0001.

used to calculate the percentage of lysogenic conversion (% lysogens) in a population (see Materials and Methods).

To validate this method, we generated a lysogenic conversion standard curve by combining known amounts of naïve RIMD cells with a stable RIMD φVP882 lysogen at ratios from 0% lysogens to 100% lysogens. Both strains carried P$_{bad}$-$q$. Almost no lysis was detected in the mixed cultures when grown in the absence of $q$-induction (S2A Fig), indicating that the background level of lytic induction in our experiments is low. Following $q$-induction, a stepwise increase in cell death occurred that was proportional to the initial ratio of lysogens:naïve cells in each mixed culture (S2B Fig). The cell death data were used to calculate the percentage of lysogens within each population, and those values agreed with the known ratios of lysogens: naïve cells dispensed into each mixture (S2E Fig). To ensure that the method worked reliably for each QS mutant, we performed the $q$-induction assay with a fully lysogenized population of each of our strains (S2F Fig). Absent $q$-induction, only low-level lysis could be detected in each mutant. By contrast, $q$-induction led to near total lysis (corresponding to 93–98% lysogenic conversion) in all mutants, verifying the $q$-induction assay is unaffected by strain genotype. Finally, we validated the $q$-induction assay against a standard curve of populations containing known quantities of lysogenic cells that we generated by qPCR amplification of a segment of the phage genome. Such qPCR analysis represents an established method for quantifying lysogens ([46]; and S2G–S2I Fig). In our case, the linear φVP882 prophage genome contains a $cos$ site that is not present when the genome is in the capsid-packaged configuration (S2G and S2H Fig). Thus, qPCR amplification of the $cos$ region distinguishes φVP882 lysogens from φVP882 phage particles. The qPCR results were comparable to those produced by $q$-induction (slopes of 0.98 and 0.86, respectively) (S2E and S2I Fig). The qPCR results confirm that $q$-induction delivers an accurate method to calculate the level of lysogenic conversion in a population and it greatly increases the throughput of our analyses.

## LuxO phosphorylation and high MOI are necessary for φVP882 lysogenic conversion of planktonic RIMD host cells

Using the above methods to quantify φVP882 lytic replication and lysogenic conversion, we investigated whether the LuxO-OpaR QS system regulates φVP882 infection outcomes in planktonic RIMD cells. We first explored whether φVP882 infection outcomes depend on MOI. We infected WT, low-cell-density-locked $luxO^{D61E}$ and $\Delta opaR$ strains, and the high-cell-density-locked $luxO^{D61A}$ strain with φVP882 at phage:host ratios from 1:100 to 1:100,000. Fig 2B shows that after 24 h of infection, the WT and $\Delta opaR$ strains underwent MOI-dependent lysogenic conversion, with high lysogenic conversion at high MOIs (56% and 86%, respectively) that progressively decreased as MOI decreased. The high-cell-density-locked $luxO^{D61A}$ strain underwent significantly lower lysogenic conversion at each MOI. The low-cell-density-locked $luxO^{D61E}$ strain, however, showed high levels (>94%) of lysogenic conversion

independent of MOI. We investigated whether the differences in lysogenic conversion among the strains could be attributed to differences in φVP882 adsorption. While φVP882 adsorbs to each strain, stronger adsorption occurs to the low-cell-density-locked mutants than to the WT and high-cell-density-locked $luxO^{D61A}$ strain (S3A Fig). Adsorption differences cannot, however, explain the different patterns of lysogenic conversion between the $luxO^{D61E}$ and $\Delta opaR$ low-cell-density-locked strains, as the $luxO^{D61E}$ strain shows high lysogenic conversion even at low MOIs while the $\Delta opaR$ strain does not. Together, these results indicate that lysogenic conversion is regulated by MOI and the phosphorylation state of LuxO.

## The Qrr sRNAs drive φVP882 lysogenic conversion in planktonic RIMD host cells independently of OpaR

In the LuxO-OpaR QS cascade, the Qrr sRNAs lie downstream of LuxO and upstream of OpaR (Fig 1A). At low cell density, LuxO~P activates expression of the *qrr* genes, and the Qrr sRNAs repress *opaR*. One possible explanation for our above result showing a difference in the level of lysogenic conversion that occurs when LuxO is phosphorylated versus that when OpaR is absent is that, at low cell density, the Qrr sRNAs control φVP882 lysogenic conversion through some OpaR-independent mechanism. To test this notion, we infected our set of RIMD strains carrying P$_{bad}$-q with φVP882 for 24 h and measured the ensuing levels of lysis and lysogeny (Fig 2A).

Regarding lysogenic conversion, no lysogenic conversion occurred in RIMD WT, which grows to high cell density during the experiment, or in the high-cell-density-locked $luxO^{D61A}$ strain (Figs 2C and S1A). By contrast, 96% lysogenic conversion occurred in both the $luxO^{D61E}$ and $luxO^{D61E} \Delta opaR$ low-cell-density-locked strains (Fig 2C). Infection of the low-cell-density-locked $\Delta opaR$ and $luxO^{D61A} \Delta opaR$ strains, however, resulted in only 12% and 0% lysogenic conversion, respectively (Fig 2C). These results suggest that in planktonic RIMD cells, LuxO$^{D61E}$-driven constitutive production of the Qrr sRNAs causes φVP882 to undertake the lysogenic lifestyle via an OpaR-independent mechanism. Importantly, these lysogenic conversion results do not parallel the adsorption results (S3A Fig). Specifically, the low-cell-density-locked $\Delta opaR$ strain shows maximal adsorption (>99% particles adsorbed) but only undergoes 12% lysogenic conversion, while the $luxO^{D61E} \Delta opaR$ double mutant undergoes both maximal adsorption and complete lysogenic conversion. The difference in lysogenic conversion between these two strains suggests that a mechanism other than viral attachment drives the propensity for lysogenic conversion to occur. Regarding lysis, each of our test strains showed a >2,900-fold increase in production of phage particles after 24 h of infection, demonstrating that each RIMD strain can be infected and can promote φVP882 lytic replication (S3C Fig). Phage particle production across the test strains mirrored the adsorption capabilities (S3A Fig), indicating that lytic replication is proportional to the total number of attached phage particles. Basal-level spontaneous lytic induction in the $luxO^{D61E}$ and $luxO^{D61E} \Delta opaR$ strains accounts for their abilities to produce high numbers of phage particles while having a high propensity for lysogenic conversion (S3E Fig).

The results from the MOI and epistasis experiments (Fig 2B and 2C) suggest that, for φVP882-directed lysogenic conversion of RIMD to occur, a sufficiently high MOI must exist concurrent with the high-level presence of Qrr sRNAs in the host cell. During experiments with 24 h infection periods in which repeated rounds of infection occur, the MOI of φVP882 changes over time from low to high as successive rounds of lytic replication and phage particle production occur. To assess the effects of the Qrr sRNAs at discrete φVP882 MOIs, we captured the frequency of lysogenic conversion that occurs in a single round of φVP882 infection. To do this, we measured lysogenic conversion of our panel of test strains over two hours both

at low and at high MOIs. At the low MOI, lysogenic conversion could not be detected in any strain, while at the high MOI, lysogenic conversion occurred in all strains, albeit significantly more frequently in strains possessing high levels of Qrr sRNAs (i.e., in the $\Delta opaR$ and $luxO^{D61E}$ $\Delta opaR$ strains) than in the strain possessing only low levels of the Qrr sRNAs (i.e., in the $luxO^{D61A}$ $\Delta opaR$ strain) (Fig 2D). Again, these data suggest that lysogenic conversion is favored when multiple φVP882 particles infect a cell harboring high levels of Qrr sRNAs.

We reasoned that under conditions where the required MOI is met, if the Qrr sRNAs are indeed responsible for driving φVP882 lysogenic conversion of RIMD at low cell densities, then overexpression of *qrr* genes should promote lysogeny at high cell density. To test this idea, we introduced tetracycline-inducible *qrr*2 ($P_{tet}$-*qrr*2) on a vector into our test strains and examined its effect on the ability of φVP882 to lysogenize. The Qrr sRNAs share high sequence identity, and *qrr*2 is the most highly expressed of the five *qrr* genes in RIMD, which is why we selected it for this analysis (S4A Fig). Furthermore, Qrr2 alone is sufficient to repress *opaR* and drive low-cell-density behaviors in RIMD [47]. We validated Qrr2 production from our construct in *E. coli* by demonstrating its ability to bind the *opaR* mRNA 5' UTR and repress translation and in RIMD by showing its ability to repress luciferase production from an OpaR-activated $P_{luxC}$ transcriptional reporter (S4B and S4C Fig, respectively). As expected, the $luxO^{D61E}$ strain underwent near total lysogenic conversion when it carried the empty vector (EV) or the $P_{tet}$-*qrr*2 construct when uninduced (EV = 93% and $P_{tet}$-*qrr*2 = 97%) or induced with anhydrotetracycline (aTc) (EV = 93% and $P_{tet}$-*qrr*2 = 98%) (Fig 2E). Presumably, lysogenic conversion occurs in the uninduced cultures because, at low cell density, there is high-level native production of Qrr sRNAs from the chromosome, and that saturates the putative mRNA target controlling φVP882 lysogenic conversion. In the WT and $luxO^{D61A}$ strains, by contrast, minimal lysogenic conversion occurred when they carried the *qrr*2 construct in the absence of induction (WT = 20% and $luxO^{D61A}$ = 0%). Induction of *qrr*2 significantly increased the level of lysogenic conversion in the WT and the $luxO^{D61A}$ strain (WT = 75% and $luxO^{D61A}$ = 24%) (Fig 2E). The difference between the results for the WT and the $luxO^{D61A}$ mutant are likely due to the WT strain producing endogenous Qrr sRNAs during the low-cell-density growth phase compared to the lower endogenous Qrr sRNA production that occurs in the high-cell-density-locked $luxO^{D61A}$ mutant (S4A Fig). These results support our hypothesis that high-level production of Qrr sRNAs at low cell density drives φVP882 lysogenic conversion of planktonic host cells following infection. Given that OpaR is dispensable for this effect (Fig 2B and 2C), to direct lysogenic conversion, the Qrr sRNAs must target a currently unknown host or φVP882 mRNA transcript.

## Derepression of the *scrABC* operon encoding the surface-sensing system at low cell density is required for φVP882 lysogenic conversion of surface-associated RIMD cells

The RIMD LuxO-OpaR QS cascade modulates surface behaviors including biofilm formation and swarming motility [10]. At low cell density, the ScrABC surface-sensing system catalyzes degradation of the c-di-GMP pool which promotes lateral flagella production and swarming motility [14–16]. At high cell density, OpaR directly represses the *scrABC* and lateral flagellar operons, while directly activating the *cps* operon ([11,18,25–27]; and Fig 1B). Thus, OpaR drives an increase in the c-di-GMP pool, reduced lateral flagella production, and increased exopolysaccharide production, all of which promote biofilm formation. With our new knowledge that LuxO~P and high Qrr sRNA levels foster φVP882 lysogenic conversion during infection of planktonic RIMD cells, we wondered whether components of the LuxO-OpaR QS cascade, which are intimately connected to the regulation of surface behaviors, also control

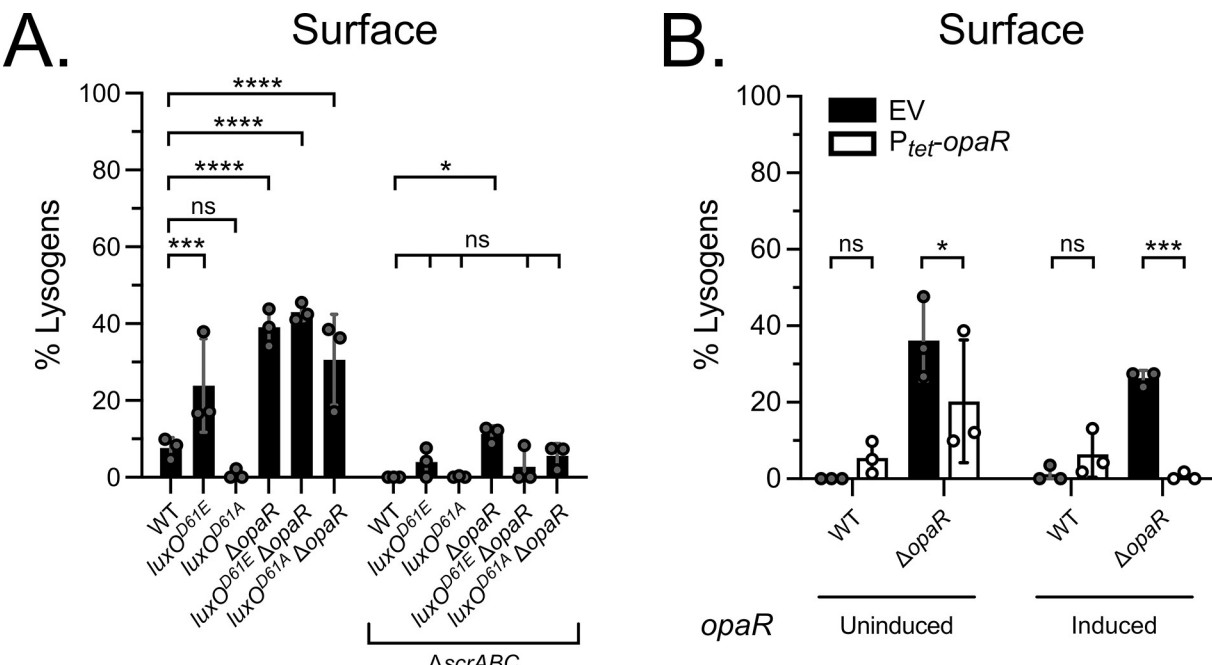

**Fig 3. OpaR suppresses φVP882 lysogenic conversion in surface-associated RIMD via repression of the surface-sensing system encoded by *scrABC*.** (A) Percent lysogens in the indicated surface-associated strains either containing or lacking the *scrABC* operon (Δ*scrABC*). (B) Percent lysogens in the indicated strains following complementation with *opaR* (EV = empty vector, P*tet*-*opaR* = inducible *opaR*) during infection on a surface. *opaR* was uninduced (-aTc) or induced (+aTc). (A,B) All experiments were performed in biological triplicate (n = 3). Symbols represent individual replicate values. Bars represent means. Error bars represent standard deviations. Results with *q*-induction (+arabinose) are shown. Significance was determined by two-way ANOVA with Tukey's multiple comparisons test to determine adjusted p-values: (A) ns = non-significant, * p = 0.0376, *** p = 0.0007, **** p < 0.0001, (B) ns = non-significant, * p = 0.0327, *** p = 0.0003.

φVP882 lysogenic conversion of RIMD when the cells are associated with a surface. For this analysis, we again used qPCR to quantify lytic replication and our P*bad*-*q* lysogeny assay to quantify lysogenic conversion, but the analyses were performed following 24 h infection of RIMD strains grown on filters that are impermeable to both bacteria and phage particles on 1.5% agar plates. Importantly, we validated that the *q*-induction assay functions reliably for cells harvested from filters on agar surfaces (S2C–S2F Fig). Similar to planktonic cells, φVP882 adsorbs to all of the test strains, albeit with less affinity to WT and *luxO^D61A* strains than to the low-cell-density-locked strains (S3B Fig).

Levels of φVP882 lysogenic conversion of surface-associated RIMD strains were generally lower than those in infections carried out in liquid (Fig 3A). Strikingly, however, the pattern of φVP882 lysogenic conversion across the surface-associated RIMD strains was different from that in planktonic cells. Analogous to what occurred in planktonic infections, only low-level lysogenic conversion occurred in surface-associated WT and high-cell-density-locked *luxO^D61A* RIMD cells following φVP882 infection (8% and <1%, respectively; Fig 3A). However, unlike in infections carried out in liquid, all the Δ*opaR* strains underwent significant lysogenic conversion when infected on a surface (39%, 43%, and 31% for Δ*opaR*, *luxO^D61E* Δ*opaR*, and *luxO^D61A* Δ*opaR*, respectively; Fig 3A). The *luxO^D61E* strain, in which *opaR* is repressed by the Qrr sRNAs but not absent, showed a lower but still significant increase in lysogenic conversion (24%) compared to WT RIMD (Fig 3A). Unlike what we showed for planktonic cells (Figs 2C and S3A), lysogenic conversion in surface-associated cells correlates with adsorption affinity (Figs 3A and S3B), indicating that attachment to the host cell may be a driving feature on a surface. As with the planktonic strains, we verified that all strains could be infected by φVP882 and could

produce new viral particles by lytic replication (S3D and S3F Fig). Higher overall viral particle production occurred in surface-associated strains compared to what occurred during planktonic infections (S3D Fig), indicating a possible preference of φVP882 for lysis over lysogeny during infection on a surface (Fig 3A).

Collectively, the data indicate that, unlike during infection of RIMD in liquid where OpaR-independent Qrr sRNA activity controls lysogenic conversion, OpaR regulates the establishment of lysogeny on a surface. Specifically, on surfaces, the absence of OpaR leads to increased φVP882 lysogenic conversion, and therefore, OpaR must repress φVP882 lysogenic conversion. Indeed, complementation of the ΔopaR strain with tetracycline-inducible opaR ($P_{tet}$-opaR) led to a complete loss of lysogenic conversion on the surface (Fig 3B). We verified that opaR was expressed using $P_{luxC}$ and $P_{cpsA}$ transcriptional reporters, both of which are activated by OpaR (S5A and S5B Fig, respectively). These results demonstrate that φVP882 lysogenic conversion is controlled by distinct QS regulators depending on whether the host cells are infected during planktonic or surface-associated growth.

A key difference between planktonic and surface-grown RIMD strains is that increased expression of the scrABC operon encoding the ScrABC surface-sensing system occurs during low-cell-density surface growth ([10,48]; and S6A Fig). The ScrABC system is activated by mechanical cues upon surface association and is further modulated by cell density via OpaR regulation (Figs 1B and S6A). These features make ScrABC a good candidate to confer context-dependent regulation of φVP882 lysogenic conversion of RIMD. To investigate this notion, we deleted scrABC from each of our test strains. As above, we infected the strains with φVP882 on filters on agar surfaces. Deletion of the scrABC operon led to a large reduction in lysogenic conversion in each test strain (Fig 3A), but it did not impair φVP882 adsorption to cells (S3B Fig). This finding indicates that, analogous to planktonic cell infection outcomes, aspects other than φVP882 attachment to the host mediate lysogenic conversion in surface-associated cells. These results suggest that the de-repression of the scrABC operon that occurs in the absence of OpaR at low cell density enables φVP882 to establish lysogeny during surface infection.

## Degradation of the global c-di-GMP pool leads to increased φVP882 lysogenic conversion of RIMD on a surface

The ScrABC surface-sensing system controls surface behaviors through modulation of the intracellular c-di-GMP pool. When scrABC is expressed in RIMD cells on a surface at low cell density (S6A Fig), ScrC behaves as a phosphodiesterase and degrades c-di-GMP [15]. To verify this activity, we used a fluorescent reporter [49] to assess c-di-GMP abundance across planktonic and surface-associated RIMD strains. Irrespective of growth condition, all ΔopaR strains possessed less c-di-GMP than when OpaR was present, and the reductions in c-di-GMP abundance were more dramatic in surface-associated RIMD compared to planktonic cells (S6B Fig). Together, these results are consistent with activation of scrABC expression and phosphodiesterase activity in ΔopaR strains following surface association.

We assessed whether scrABC-dependent degradation of the c-di-GMP pool is sufficient to trigger φVP882 lysogenic conversion in RIMD. We ensured that we could produce decreases and increases, respectively, in c-di-GMP abundance using tetracycline-inducible constructs to overexpress scrABC ($P_{tet}$-scrABC), harboring the WT scrC in which ScrC functions as a phosphodiesterase, or a mutant ScrC ($P_{tet}$-scrC$^{E554A}$) that functions exclusively as a diguanylate cyclase ([15]; and S6C Fig). We verified that ScrABC and the mutant ScrC$^{E554A}$ were produced using a transcriptional reporter for the lateral flagellin ($P_{lafA}$), which is highly expressed during swarming due to ScrABC-driven decreases in c-di-GMP abundance and is repressed by

ScrC$^{E554A}$-driven increases in c-di-GMP (S6D and S6E Fig, respectively). With this strategy in hand, we once again infected the Δ*scrABC* and Δ*opaR* strains on agar surfaces (Fig 4A). As a reminder, on a surface, the Δ*scrABC* strain possesses high c-di-GMP and undergoes minimal lysogenic conversion, while the Δ*opaR* strain possesses low c-di-GMP and undergoes high-level lysogenic conversion (Fig 3A). Thus, these two strains allow us to maximally follow increases (Δ*scrABC*) and decreases (Δ*opaR*) in lysogenic conversion in response to alterations in c-di-GMP abundance. Complementation of the Δ*scrABC* strain with P$_{tet}$-*scrABC* restored lysogenic conversion to levels exceeding that in the Δ*opaR* strain (Fig 4A; EV = 0% and P$_{tet}$-*scrABC* = 58%). By contrast, overexpression of *scrC*$^{E554A}$ resulted in near complete loss of lysogenic conversion in the Δ*opaR* strain (Fig 4A; EV = 30% and P$_{tet}$-*scrC*$^{E554A}$ = 2%). Together, these results support QS- and ScrABC-dependent alterations to the c-di-GMP pool as key to driving φVP882 lysogenic conversion of surface-associated RIMD cells. Importantly, lysogenic conversion during planktonic infection was unaffected by deletion of *scrABC*. Specifically, in liquid, both the *luxO*$^{D61E}$ and *luxO*$^{D61E}$ Δ*scrABC* mutants underwent near total lysogenic conversion (96% and 97%, respectively; Fig 4B). This result is consistent with our finding and previously reported data showing that *scrABC* is repressed in planktonic culture ([48]; and S6A and S6B Fig). Thus, c-di-GMP-driven modulation of φVP882 lysogenic conversion is specific to surface-associated host cells.

c-di-GMP binding effector proteins can respond to changes in the global c-di-GMP pool and/or to changes in local c-di-GMP abundance driven by specific diguanylate cyclases and phosphodiesterases [50,51]. To test whether the c-di-GMP responsive factor(s) driving φVP882 lysogenic conversion of surface-associated RIMD strains respond to a global reduction in the c-di-GMP pool or whether they respond specifically to changes driven by ScrC phosphodiesterase activity, we performed our surface infection experiments following modulation of c-di-GMP levels using the RIMD phosphodiesterase TpdA and the RIMD diguanylate cyclase GefA. We constructed P$_{tet}$-*tpdA* and P$_{tet}$-*gefA* and verified their activities by measuring their effects on c-di-GMP abundance and lateral flagellin expression. As expected, production of TpdA and GefA had opposite effects; TpdA reduced the c-di-GMP pool and increased *lafA* expression, while GefA increased the c-di-GMP pool and reduced *lafA* expression (S6F–S6H Fig). Fig 4C shows that overexpression of *tpdA* in the Δ*scrABC* strain restored lysogenic conversion (EV = 21% and P$_{tet}$-*tpdA* = 75%). By contrast, overexpression of *gefA* resulted in the loss of lysogenic conversion in the Δ*opaR* strain (EV = 75% and P$_{tet}$-*gefA* = 22%). Thus, reductions in the global c-di-GMP pool, that need not be exclusively catalyzed by the ScrC phosphodiesterase, promote lysogenic conversion of RIMD on surfaces.

## Discussion

In this report, we identified *V. parahaemolyticus* sensory components that control the φVP882 lysis-lysogeny decision during infection of naïve host cells. We discovered that *V. parahaemolyticus* RIMD cells can be infected and lysed by φVP882 at all cell densities, but *V. parahaemolyticus* RIMD cells at low cell density favor lysogenic conversion. Key was our finding that the sensory components necessary to drive lysogenic conversion varied based on the physical environment of the infected host cell. Specifically, high-level Qrr sRNA production drives lysogenic conversion in low-cell-density planktonic host cells, while ScrABC-directed degradation of the global c-di-GMP pool drives lysogenic conversion in low-cell-density surface-associated host cells. The components and mechanisms connecting the Qrr sRNAs and ScrABC to φVP882 lysogenic conversion capability are currently unknown.

Our data support the following scenario: When φVP882 initially encounters a susceptible host cell, it can adsorb at any cell density, albeit with greater affinity to cells in the low-cell-

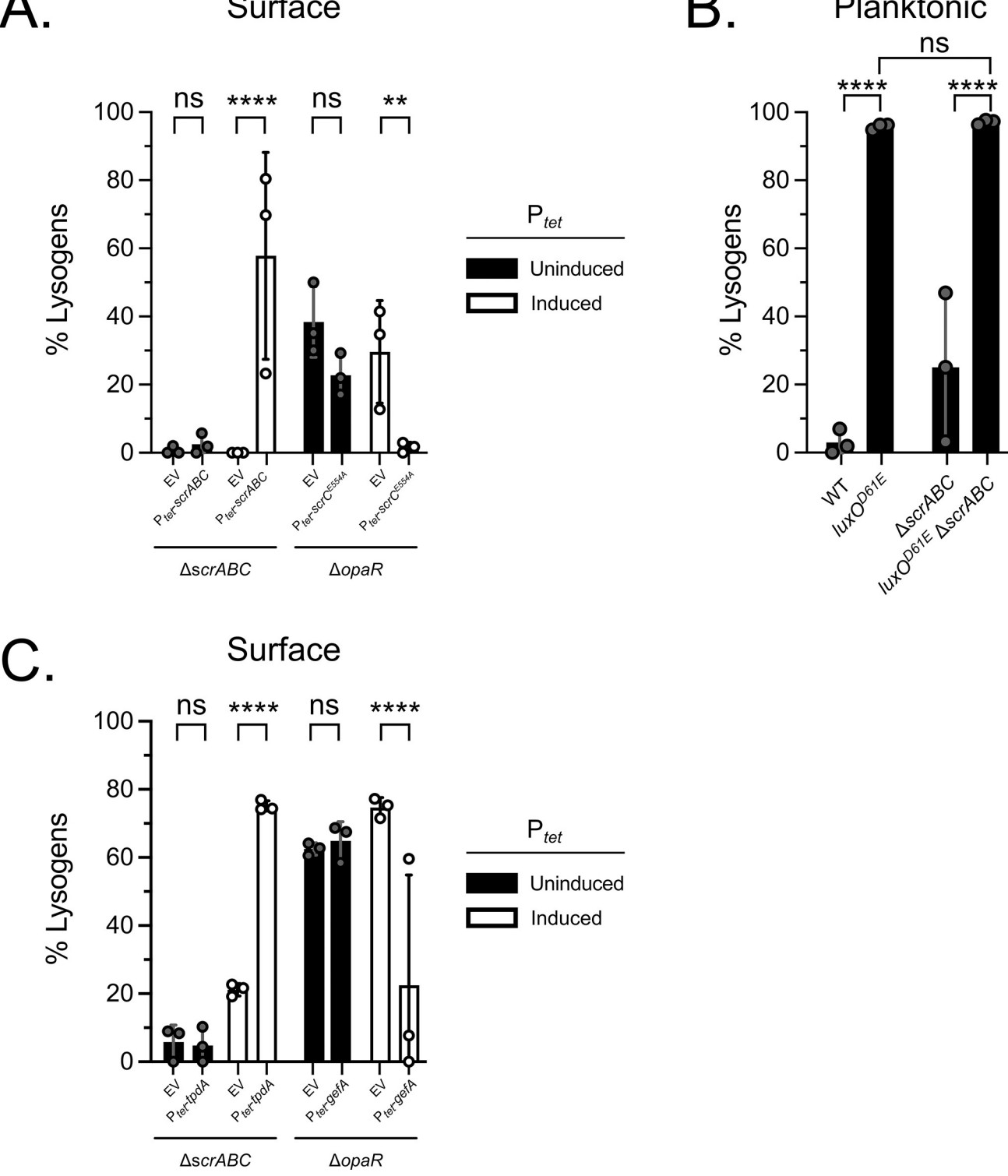

**Fig 4. φVP882 lysogenic conversion of surface-associated RIMD host cells is regulated by the global c-di-GMP pool.** (A) Percent lysogens in the indicated surface-associated strains without or with induction of *scrABC* ($P_{tet}$-*scrABC*; in this configuration ScrC functions as a phosphodiesterase) or *scrC*$^{E554A}$ ($P_{tet}$-*scrC*$^{E554A}$; a diguanylate cyclase) during infection. EV = empty vector control. (B) Percent lysogens in the indicated planktonic strains. (C) Percent lysogens in the indicated surface-associated strains without or with induction of *tpdA* ($P_{tet}$-*tpdA*; a phosphodiesterase) or *gefA* ($P_{tet}$-*gefA*; a diguanylate cyclase) during infection. EV = empty vector control. (A-C) All experiments were performed in biological triplicate (n = 3). Symbols represent individual replicate values. Bars represent means. Error bars represent standard deviations. Results with *q*-induction (+arabinose) are shown. (A,C)

Expression from plasmids was uninduced (-aTc) or induced (+aTc). (A-C) Significance was determined by two-way ANOVA with Tukey's multiple comparisons test to determine adjusted p-values: (A) ns = non-significant, ** p = 0.0054, **** p < 0.0001, (B) ns = non-significant, **** p < 0.0001, (C) ns = non-significant, **** p < 0.0001.

density QS state. After genome entry, φVP882 must lyse or lysogenize its host. Lysis is considered the default outcome for temperate phages [31], which is consistent with our results, as infected RIMD strains at all cell densities produce viral particles proportional to the level of phage adsorption. In model temperate phages such as phage λ, the primary mechanism promoting lysogeny is a high MOI because when multiple phage genomes enter the infected cell, a high dose of lysogeny-promoting phage regulatory proteins is supplied [33,34]. The discovery of phage-bacterial and phage-phage communication systems that measure extracellular small molecule signals that modulate phage lifestyle decision making, however, has added new regulatory complexity to the rather straightforward notion of MOI as the main arbiter of the lysis-lysogeny decision [39,42,43,52–54]. Our results show that in the case of φVP882, the initial lysis-lysogeny decision is controlled by both MOI and the phosphorylation state of LuxO. When RIMD cells are infected at a high MOI and at low cell density, i.e., when LuxO is phosphorylated, lysogenic conversion is preferred. During low-cell-density planktonic growth, maximal production of the Qrr sRNAs drives lysogenic conversion through an OpaR-independent mechanism (Fig 5, top left). When associated with a surface, Qrr sRNA repression of OpaR production leads to derepression of the *scrABC* operon and degradation of the global c-di-GMP pool, which promote φVP882 lysogenic conversion (Fig 5, bottom left). We expect that this mechanism involves an unknown c-di-GMP responsive effector. The Qrr sRNAs and ScrABC are both situated downstream of LuxO in the *V. parahaemolyticus* QS cascade (Fig 1), suggesting that LuxO is the master regulator of the φVP882 lysis-lysogeny decision. This idea is further bolstered when we consider data concerning φVP882 induction from lysogeny. Lysogenic φVP882 undergoes lytic induction when its VqmAφ QS receptor/transcription factor detects the autoinducer DPO at high host cell density [39]. Dephosphorylation of LuxO at high cell density leads to increased VqmAφ-dependent lytic induction ([43]; and Fig 5, right). While not investigated here, it is possible that LuxO-dependent regulation of lysogenic conversion and VqmAφ-dependent regulation of lytic induction are connected. Specifically, phosphorylation/dephosphorylation of LuxO may enable toggling between φVP882 lysogeny (via Qrr sRNAs or ScrABC, depending on the physical context of the host) and lysis (via DPO-VqmAφ) both when the initial lysis-lysogeny decision is made and later in infection when the decision to transition from lysogeny maintenance to lytic induction is undertaken (Fig 5). Alternatively, it is possible that the connection between LuxO phosphorylation/dephosphorylation and φVP882 lysis-lysogeny transitions is indirect and due to φVP882 detection of a shift in general host physiology that is triggered by the Qrr sRNAs in planktonic cells and decreased c-di-GMP in surface-associated cells.

Concerning the role of c-di-GMP, our results indicate that the φVP882 lysis-lysogeny decision is only affected by OpaR-dependent changes in the global c-di-GMP pool when RIMD cells are infected on a surface, not when they are in liquid (Fig 4A and 4B). Our results and other published work, however, have demonstrated that OpaR promotes shifts in the global c-di-GMP pool in both planktonic and surface-associated RIMD cells ([24,55]; and S6B Fig). This apparent inconsistency cannot simply be explained by ScrABC activation being restricted to surface growth because TpdA, a phosphodiesterase that is highly active in planktonic cells [24], was also capable of driving φVP882 lysogenic conversion of RIMD (Fig 4C). This finding suggests that the specific c-di-GMP responsive effector(s) that drives φVP882 lysogenic conversion might only be produced/active in surface-associated cells. The RIMD genome encodes

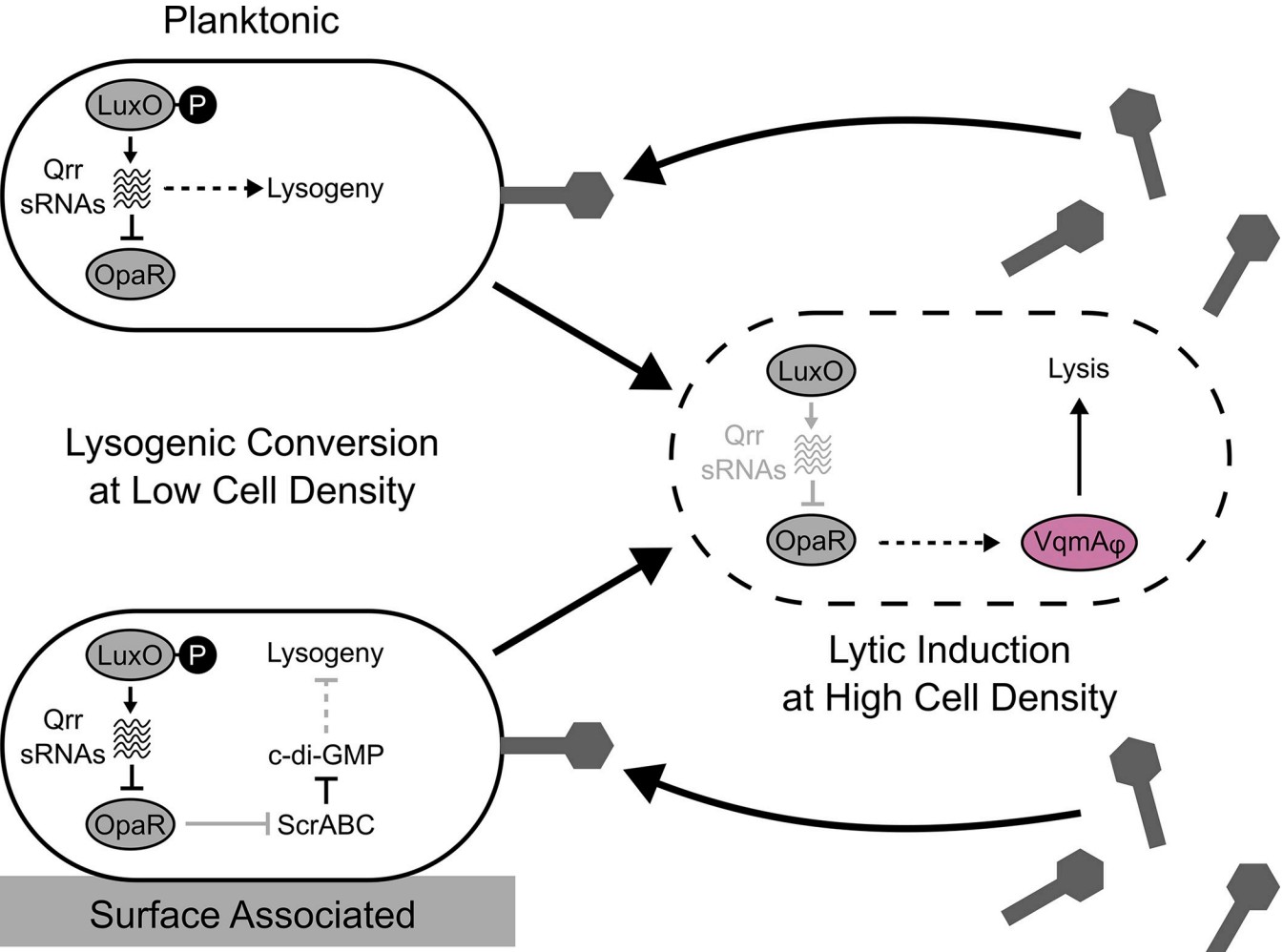

**Fig 5. Regulation of φVP882 lysis-lysogeny transitions requires different QS components depending on the physical environment of the *V. parahaemolyticus* RIMD host.** (Left) Schematics of the regulatory networks controlling φVP882 lysogenic conversion of low-cell-density host cells. (Top) In planktonic RIMD host cells, lysogenic conversion is driven by LuxO~P and the Qrr sRNAs in an OpaR-independent manner. (Bottom) In surface-associated RIMD host cells, LuxO~P drives derepression of *scrABC* and degradation of the c-di-GMP pool, promoting lysogenic conversion. (Right) Schematic of the known relationship between LuxO and the φVP882-encoded transcription factor and QS receptor VqmAφ [39,43]. At high cell density, QS promotes lytic induction through a VqmAφ-dependent mechanism. (Left, Right) Solid and dashed arrows indicate direct and indirect regulation, respectively. Black and gray arrows indicate active and inactive regulatory pathways, respectively, under the specified physical conditions. Large arrows and phage particles indicate the φVP882 lifecycle.

many known and putative c-di-GMP responsive effectors, of which three have been shown to control surface phenotypes–ScrO, CpsS, and CpsQ [18,19,56], making them potential candidates for future exploration of host factors involved in φVP882 lysogenic conversion on a surface. In an analogous vein, high-level Qrr sRNA production was not sufficient to drive lysogenic conversion in surface-associated cells in the absence of ScrABC (Fig 3A). Following the same logic as above, perhaps the Qrr sRNAs only drive lysogenic conversion in planktonic cells because their downstream target is a planktonic-specific factor(s). Possessing surface growth- and planktonic growth-specific intermediate factors could provide RIMD an effective mechanism to confine the influences of the Qrr sRNAs and c-di-GMP abundance to their respective physical contexts despite both the Qrr sRNAs and c-di-GMP having other roles in planktonic and surface-associated cells [22,24,47].

Previously, we hypothesized that φVP882 integrates host cell density information into its lysis-lysogeny transitions to ensure lytic replication is favored under conditions that maximize transmission to new host cells [39,43]. The preference for lysogeny over lysis in host cells at low cell density aligns with this maximum transmission hypothesis. Indeed, we show here that lysogeny occurs in the presence of high Qrr sRNA levels and low c-di-GMP abundance, both of which coincide with maximal phosphorylation of LuxO and the absence of OpaR, i.e., low-cell-density conditions. Germane to the present work is that, at low cell density, when OpaR levels are at their lowest, *V. parahaemolyticus* cells are more motile than at high cell density when OpaR is present and is functioning to, respectively, directly and indirectly repress the polar (swimming) and lateral flagellar systems (surface-associated swarming) [9,10,47,57,58]. Perhaps lysogenizing highly motile cells allows φVP882 to disperse its genome over greater distances than could be achieved by virion particles released from lysed non-motile cells. Phage "hitchhiking" has been demonstrated but is typically described as phage capsids non-specifically binding to membrane components on non-host cells and being transported to locales in which legitimate host cells reside, enabling phage infection and replication [59,60]. The φVP882 low-cell-density lysogeny mechanism we have uncovered here could be a form of "hitchhiking" that allows φVP882 prophages to disseminate and exist in a dormant state until a high density of host cells is achieved. Following dephosphorylation of LuxO and production of OpaR in such a high-density population, motility is suppressed, VqmAφ is produced, and the switch to lytic induction occurs.

Studies investigating connections between phage infection and QS have primarily focused on QS regulation of host defenses against phage infection [61–69]. QS control of phage lysis-lysogeny transitions is a new area of study. As additional QS-responsive temperate phages are discovered [39–42,70–72], probing lysis-lysogeny transition dynamics in the context of host QS-controlled group behaviors could provide insight into the evolution of cross-domain phage-bacterial QS interactions and the costs and benefits to each entity under particular environmental conditions.

## Materials and methods

### Bacterial strains, reagents, and growth conditions

*E. coli* strains were grown with aeration in Lysogeny Broth (LB-Miller, BD-Difco) at 37˚C. *V. parahaemolyticus* RIMD strains were grown with aeration in LB or LB with 2% NaCl at 30˚C. Strains used in the study are listed in S1 Table. Unless otherwise noted, antibiotics, were used at: 100 µg mL$^{-1}$ ampicillin (Amp, Sigma), 50 µg mL$^{-1}$ kanamycin (Kan, GoldBio), 50 µg mL$^{-1}$ polymyxin B (Pb, Sigma), 5 µg mL$^{-1}$ chloramphenicol (Cm, Sigma), and 15 µg mL$^{-1}$ gentamycin (Gm, Sigma). L-arabinose (Sigma) and L-dextrose (Sigma) were supplied at a final concentration of 0.2%. Anhydrotetracycline (aTc, Takara Bio) was supplied at a final concentration of 100 ng mL$^{-1}$.

### Cloning techniques

All primers used for plasmid construction and qPCR, listed in S2 Table, were obtained from Integrated DNA Technologies. FastCloning was employed for plasmid assembly [73]. Briefly, PCR with iProof polymerase (Bio-Rad) was used to generate cloning inserts and linear plasmid backbone DNA. In cases in which inserts could not be generated by PCR, fragments were synthesized by Integrated DNA Technologies. In the case of pRE112, linear plasmid backbone was generated by restriction digestion with SmaI (NEB). Plasmid backbone DNA was treated with DpnI (NEB) to remove PCR template DNA. Cloning inserts and linear plasmid backbones were added to chemically competent *E. coli* cells, and plasmid assembly was carried out

by the transformed cells. All assembled plasmids were verified by Sanger sequencing. Plasmids used in this study are listed in S3 Table. Transfer of plasmids into *V. parahaemolyticus* RIMD strains was carried out by conjugation followed by selective plating on LB plates supplemented with appropriate antibiotics. Point mutations and deletion mutants were generated by allelic exchange with sucrose counterselection. Mutations were verified by colony PCR and Sanger sequencing.

## Phage purification, concentration, and titering

An overnight culture of *V. parahaemolyticus* strain 882 (the original strain from which φVP882 was first isolated) carrying φVP882 and arabinose-inducible *vqmAφ* ($P_{bad}$-*vqmAφ*) on a vector was diluted 1:100 into fresh LB with 2% NaCl and grown with aeration at 30˚C until the culture reached mid-logarithmic growth ($OD_{600}$ = 0.2–0.4). At this point, arabinose was added to induce φVP882 lytic replication. Lysis was tracked by $OD_{600}$ until the culture completely cleared. Cellular debris was removed from the lysate by centrifugation (3000 rpm, 15 min). The supernatant was collected and treated with DNase (Roche) and RNase (Roche) at 5 μg mL$^{-1}$, each, for 1 h at room temperature, followed by filtration through a 0.22 μm bottle top vacuum filter (Millipore). The filtrate was treated with 3% v/v Tween 80 and 35% m/v ammonium sulfate and was subjected to centrifugation (3000 xg, 10 min, 4˚C). The pellicle was collected and resuspended in SM buffer (100 mM NaCl, 8 mM $MgSO_4$•$7H_2O$, 50 mM Tris-Cl). Phage stocks were titered by qPCR. Briefly, an aliquot of phage stock was treated with RQ1 DNase (Promega) to remove remaining free DNA, leaving only encapsidated phage genomes. Phage capsids were lysed during the DNase heat denaturation step, and φVP882 genomes were quantified by qPCR with primers against the *gp69* gene. Full degradation of free host genomic DNA by the DNase treatment was verified with primers against the host *ompW* gene. Copy number was determined by absolute quantitation against a standard curve of plasmid DNA harboring the *gp69* gene. Viable phage particles were quantified by plaque assay. *V. parahaemolyticus* RIMD host cells were suspended in molten LB medium containing 0.3% top agar and 10 mM $CaCl_2$. Suspensions were overlayed onto plates containing LB medium with 1.5% agar. Ten-fold serial dilutions of phage stock ($10^0$ to $10^{-7}$) were spotted on the solidified top agar overlay. Plates were incubated for 18 h at 30˚C to allow plaque formation. Plaques were counted to calculate plaque forming units (PFU mL$^{-1}$).

## Quantitation of φVP882 adsorption, lytic replication, and lysogenic conversion during planktonic and surface-associated infections

To test φVP882 adsorption, overnight cultures of *V. parahaemolyticus* RIMD strains carrying an arabinose-inducible copy of the φVP882 lytic regulator *q* ($P_{bad}$-*q*) were diluted 1:100 in fresh LB medium containing 10 mM $CaCl_2$ and Kan. For planktonic cell adsorption assays, diluted cultures were grown with aeration at 30˚C for 4 h. Approximately $10^8$ φVP882 particles were added to 250 μL of each culture. Phage-treated cultures and cell-free, phage-only controls were incubated without shaking at 30˚C for 10 min. The samples were gently pelleted (10 min, 500 xg, 4˚C) to remove cells and adsorbed phage particles. For surface-associated cell adsorption assays, 25 μL of each diluted culture was spotted onto a 0.025 μm mixed cellulose ester (MCE) filter disc (Millipore) placed on a dry agar surface (LB with 1.5% agar and Kan), and the samples were incubated at 30˚C for 18 h. *E. coli* TOP10 cells were spotted onto MCE filter discs placed on a dry agar surface without Kan as the unadsorbed control. Filters were collected and suspended in 500 μL of LB medium containing 10 mM $CaCl_2$ and approximately $10^6$ φVP882 particles. These samples were subjected to vortex to remove the cells from the filters and allow them to become mixed with the phage particles. The cells thus treated were

incubated at 30˚C for 20 min. Samples were gently pelleted (10 min, 500 xg, 4˚C) to remove any cells and adsorbed phage particles. For both the planktonic cell and surface-associated cell samples, cell-free culture fluids were collected, filter-sterilized through 96-well 0.22 μm filter plates (Millipore), and qPCR was performed against the φVP882 *gp69* gene as described above to quantify unadsorbed phage particles.

For infection of planktonic cells, overnight cultures of *V. parahaemolyticus* RIMD strains carrying P*bad*-*q* were diluted 1:100 in fresh LB containing 10 mM $CaCl_2$ and Kan. These diluted cultures were treated with either φVP882 (MOIs are indicated in figure legends) or an equivalent volume of SM buffer. Cultures were grown with aeration at 30˚C for 24 h. Cells were collected and diluted 1:100 in fresh LB with Kan but lacking $CaCl_2$, which prevents subsequent rounds of infection by phage particles produced due to Q-driven lytic induction. For infection of surface-associated cells, immediately following treatment exactly as described for planktonic cells, 25 μL of the cultures were spotted onto 0.025 μm MCE filter discs on a dry agar surface (LB with 1.5% agar and Kan). Plates were incubated at 30˚C for 24 h. Filter disks were removed from the agar plates with sterile tweezers and transferred into 1 mL LB with Kan followed by vortex for 15 sec to dislodge cells from filters. The suspensions were diluted 1:100 in fresh LB with Kan, again lacking $CaCl_2$. When needed to maintain plasmids and to induce gene expression, Cm and aTc were added to liquid media (planktonic infections) or to agar plates (surface infections) from the start of the infection period.

To quantify φVP882 viral particle production following both planktonic and surface-associated infections, cell-free culture fluids (0.22 μm filter-sterilized) were collected at T = 0 and T = 24 h of infection. For surface infections, cell-free culture fluids were obtained from the initial cell suspensions prior to spotting on the MCE filters (T0) and the post-infection cell suspensions generated by vortex of the samples collected from the filters (T24). As described above, culture fluids were treated with DNase, and encapsidated phage genomes were quantified by qPCR against the φVP882 *gp69* gene.

Lysogenic conversion was quantified by *q*-induction. In all cases, after 24 h of infection, collection, and dilution, aliquots of cultures were transferred in duplicate to a 96-well plate. Plates were incubated at 30˚C in a BioTek Synergy Neo2 Multi-Mode plate reader. Every 5 min, the plate was shaken orbitally, and $OD_{600}$ was measured. When cultures reached mid-logarithmic growth, L-arabinose (*q*-induction) or L-dextrose (*q*-repression) was added. Plates were returned to the plate reader and $OD_{600}$ was measured every 5 min for the next 12 h. The measured $OD_{600}$ values were used to quantify the percentage of lysogenic conversion (% lysogens) in each population. To quantitate lysogenic conversion after short periods of infection (0–2 h), qPCR was performed against the φVP882 *cos* site and the host *ompW* gene. The number of phage genomes (*cos* site) and host genomes (*ompW*) were determined by absolute quantitation against appropriate standard curves. Because the φVP882 prophage is a multi-copy element, all *cos* site counts from infected samples were normalized to the *cos* site copy number per cell for a stable φVP882 lysogen grown under the same conditions as the infected strain under study.

## Calculation of lysogenic conversion from Q-driven cell lysis

Treatment with phage or SM buffer during the infection period and subsequent addition of L-arabinose or L-dextrose resulted in four experimental conditions: SM buffer/+dextrose, phage/+dextrose, SM buffer/+arabinose, phage/+arabinose. The following analysis was performed for the arabinose-treated (*q* induction) samples and the dextrose-treated (*q* repression) samples. The level of lysogenic conversion (% lysogens) was calculated for the arabinose-treated and dextrose-treated samples using their respective phage-treated and buffer-treated

conditions and the following equation:

$$\%L = \left(\frac{OD_P - (OD_V/E)}{OD_P}\right)*100$$

Here, $OD_P$ and $OD_V$ are the optical densities of the phage-treated cultures at the peak (pre-lysis) and valley (post-lysis) of the growth curve. The peak and valley of each growth curve were determined by setting a boundary at 4 h for planktonic infections and at 5 h for surface-associated infections. $OD_P$ is the maximum $OD_{600}$ value before the boundary time and $OD_V$ is the minimum $OD_{600}$ value after the boundary time. $OD_V$ is divided by an expansion factor calculated from the SM buffer-treated culture. The expansion factor can be represented by the ratio of the $OD_{600}$ of the SM buffer-treated culture at the timepoints corresponding to the peak ($N_{tP}$) and valley ($N_{tV}$) of the infected culture. The expansion factor adjusts the infected culture $OD_V$ value to correct for continued growth of the non-lysogenized cells in that culture. We assume that induction of $q$ in the infected culture leads to complete lysis of all lysogenized cells. As the lysogenized cells lyse, the growth rate of the non-lysogenized cells in that culture likely increases due to decreased culture cell density and increased nutrient availability. Thus, when growth of the phage-treated culture is reduced compared to the SM buffer-treated culture upon $q$-induction due to death of lysogenized cells in the former ($N_{tV} / N_{tP} > OD_V / OD_P$), the expansion factor is scaled by the ratio of $OD_V$ to $N_{tV}$ or the fold-reduction in $OD_{600}$ between the SM buffer-treated culture and the phage-treated culture.

$$\%L = \begin{cases} \left(\dfrac{OD_P - \left(OD_V/\left(\left(\dfrac{N_{tV}}{N_{tP}}\right)\right)\right)}{OD_P}\right)*100, \text{for } N_{tV}/N_{tP} \leq OD_V/OD_P \\[4ex] \left(\dfrac{OD_P - \left(OD_V/\left(\left(\dfrac{N_{tV}}{N_{tP}}\right) + \left(\left(\dfrac{N_{tV}}{OD_V}\right) - 1\right)\right)\right)}{OD_P}\right)*100, \text{otherwise} \end{cases}$$

$N_{tP}$ and $N_{tV}$ are calculated with a basic form of the logistic equation commonly used in ecology and evolution:

$$N_t = \frac{K}{1 + \left(\frac{K-N_0}{N_0}\right)e^{-rt}}$$

Here, $N_0$ represents the population size at the beginning of the growth curve. $K$ represents the carrying capacity of the culture. The intrinsic growth rate of the population is given by $r$. Finally, $t$ represents time. These parameters are determined by fitting the above logistic equation to the experimental growth curves of the SM buffer-treated cultures using the R package growthcurver (v 0.3.1) [74].

## Reporter assays

Overnight cultures of *V. parahaemolyticus* RIMD strains carrying bioluminescent or fluorescent reporter constructs were diluted 1:100 in fresh LB with appropriate antibiotics except for strains carrying the P*luxC*-*luxCDABE* construct, which were diluted 1:1000. Overnight cultures of *E. coli* strains carrying fluorescent reporter constructs were diluted 1:100 in fresh M9-glycerol with appropriate antibiotics. In the case of planktonic cell assays, diluted cultures were dispensed (200 μL) into 96-well plates (Corning Costar). For surface-associated cell assays, diluted cultures

were spotted (3 μL) on solidified 1.5% LB agar (100 μL) with appropriate antibiotics in 96-well plates. The liquid and agar media were supplemented with aTc, arabinose, or dextrose where specified. Optical density, fluorescence, and bioluminescence were measured with a BioTek Synergy Neo2 Multi-Mode plate reader. In the case of *luxCDABE*-based transcriptional reporters, relative light units (RLU) were calculated by dividing bioluminescence by optical density (for planktonic cell assays) or by dividing bioluminescence by constitutive mScarlet-I signal from the host chromosome (for surface-associated cell assays). If an experiment required reporter expression to be compared between planktonic and surface conditions, bioluminescence was normalized to constitutive mScarlet-I signal under both conditions. In the case of fluorescence-based translational reporters in *E. coli*, relative fluorescence units (RFU) were calculated by dividing GFP signal by optical density. Regarding the fluorescence-based c-di-GMP biosensor in *V. parahaemolyticus* RIMD strains, RFU were calculated by dividing the TurboRFP reporter signal by the constitutive AmCyan signal. Both fluorescent reporters are encoded on the same vector [49].

## Supporting information

**S1 Table. Strains used in this study.**
(DOCX)

**S2 Table. Primers and synthesized fragments used in this study.**
(DOCX)

**S3 Table. Plasmids used in this study.**
(DOCX)

**S1 Data. Data points used to make plots that appear in this study.**
(XLSX)

**S1 Fig. Phenotypic verification of RIMD QS mutants.** (A) Assessment of QS phenotypes in the indicated strains carrying a QS-activated *lux* reporter ($P_{luxC}$-*luxCDABE*). Relative Light Units (RLU) are bioluminescence normalized to $OD_{600}$. All experiments were performed in biological triplicate (n = 3). Lines and symbols represent the means and shaded areas represent the standard deviations. (B) Representative stereoscope images of swarming morphologies of the indicated strains 12 h post-inoculation. The bottom right corner of each image depicts the center of the colony. White arrows indicate the outer edges of the swarm flares in the WT and low-cell-density-locked strains. Swarming radius is measured from the bottom right corner of each image to the edge of the swarm flare. (C) Representative stereoscope images of biofilm morphologies for the indicated strains 42 h post-inoculation. (B,C) Scale bars = 3 mm. Swarming and biofilm phenotypes for the strains mirror those reported in the literature [10].
(TIFF)

**S2 Fig. Q-driven host cell lysis provides an accurate measure of lysogenic conversion in all test RIMD strains carrying φVP882.** (A-D) Growth curves for WT RIMD populations consisting of known quantities of φVP882 lysogens and naïve host cells either (A,C) without *q*-induction (+dextrose) or (B,D) with *q*-induction (+arabinose). Strains were grown planktonically (A,B) or on a surface (C,D) prior to harvesting for the *q*-induction assay. Optical densities of samples in panels B and D at the peaks and valleys of their respective growth curves were used to calculate the lysogenized portion of the population (see Materials and Methods for a detailed explanation of the calculation). (E) Standard curves of the experimentally calculated percent lysogens versus the known percent lysogens without (+dextrose, open symbols) and with (+arabinose, closed symbols) *q*-induction for planktonically- (black) or surface-grown

(red) cells. The dotted diagonal lines represent simple linear regressions performed on the induced samples. The resulting equations and R-squared values are shown. (F) Calculated percent lysogens in fully lysogenized populations of the indicated strains without (+dextrose) and with (+arabinose) *q*-induction after planktonic (left) or surface-associated (right) growth. (G) Diagram of the three genomic configurations of φVP882: the packaged linear form (top), the circular replicative form (bottom left), and the linear prophage form (bottom right). Shown are the *gp69* gene (yellow), the *cos* site (red), and the *IRS* (blue). Double-sided arrows represent the flow of genomic rearrangements. (H) Absolute quantitation of the indicated φVP882 genomic regions in naïve cells, lysogens, and purified φVP882 particles. *ompW* is a gene in the RIMD genome used to calculate host genome copy number. Dotted lines represent the limit of detection for the DNA region with the corresponding color. (I) Standard curve of percent lysogens calculated by qPCR using primers against the φVP882 *cos* site (red) versus the known percent lysogens. Primers targeting the φVP882 *IRS* (blue) were included to demonstrate that the circular replicative form of the φVP882 genome, which also contains an intact *cos* site, does not confound quantitation of φVP882 lysogens. The dotted diagonal lines represent simple linear regressions. The resulting equations and R-squared values are shown. (A-F,H,I) All experiments were performed in biological triplicate (n = 3). (A-E,I) Lines and symbols represent the means, and shaded areas represent the standard deviations. (F,H) Symbols represent individual replicate values. Bars represent means. Error bars represent standard deviations.
(TIFF)

**S3 Fig. φVP882 can adsorb to, infect, and undergo lytic replication in RIMD and all QS mutant strains in liquid and on a surface.** (A,B) φVP882 adsorption to the indicated RIMD strains shown as the percentage of phage particles removed by the cells from the culture medium after growth (A) in liquid or (B) on a surface. In A,B data are shown as 100%-% recovered phage particles. (C,D) φVP882 viral particle production in the indicated strains after infection (C) in liquid or (D) on a surface. Data are shown as the fold-change in harvested free viral particles at the end (24 h) of the infection compared to that at the beginning (0 h). (E,F) Quantitation of phage particles produced spontaneously from the indicated lysogenic strains when grown (E) in liquid or (F) on a surface. Dotted lines indicate the limit of detection by qPCR. (A-F) All experiments were performed in biological triplicate (n = 3). Symbols represent individual replicate values. Bars represent means. Error bars represent standard deviations.
(TIFF)

**S4 Fig. The *qrr* sRNA genes are expressed, and the Qrr sRNAs are functional in *E. coli* and in RIMD.** (A) Light production from transcriptional reporters of the five Qrr sRNA promoters (P$_{qrr1-5}$-*luxCDABE*) in WT RIMD and the high-cell-density-locked *luxO*$^{D61A}$ strain during planktonic growth. (B) Fluorescence output is shown from *E. coli* strains carrying an *opaR-5'UTR-gfp* translational reporter (P$_{bad}$-*opaR-5'UTR-gfp*) and either an empty vector control (EV) or a *qrr2* overexpression construct (P$_{tet}$-*qrr2*). The *opaR-5'UTR-gfp* translational reporter was uninduced (+dextrose) or induced (+arabinose) in the absence (-aTc) or presence (+aTc) of *qrr2* overexpression. Data (% RFU) are represented as percent GFP signal for each sample compared to the sample following induction of only the *opaR-5'UTR-gfp* translational reporter. (C) Light production from a transcriptional reporter of a QS-activated promoter (P$_{luxC}$-*luxCDABE*) in RIMD carrying either an empty vector control (EV) or a *qrr2* overexpression construct (P$_{tet}$-*qrr2*) is shown. Samples were uninduced (-aTc) or induced (+aTc). (A-C) All experiments were performed in biological triplicate (n = 3). Symbols represent individual replicate values. Bars represent means. Error bars represent standard deviations. (A,C) RLU are bioluminescence normalized to OD$_{600}$. (B,C) Significance was determined by two-

way ANOVA with Tukey's multiple comparisons test to determine adjusted p-values: (B) ns = non-significant, **** p <0.0001, (C) ns = non-significant, * p = 0.0101, ** p = 0.0098, *** p = 0.0002.
(TIFF)

**S5 Fig. OpaR produced from a plasmid functions in RIMD and can control high-cell-density behaviors in both planktonic and surface cultures.** (A) Light production from a transcriptional reporter of the QS-activated luciferase operon ($P_{luxC}$-$luxCDABE$) in planktonic RIMD strains carrying either an empty vector control (EV) or an $opaR$ overexpression construct ($P_{tet}$-$opaR$). RLU are bioluminescence normalized to $OD_{600}$. (B) Light production from a transcriptional reporter of the QS-activated exopolysaccharide operon ($P_{cpsA}$-$luxCDABE$) in surface-associated RIMD strains carrying either EV or $P_{tet}$-$opaR$. RLU are bioluminescence normalized to constitutive mScarlet-I signal. (A,B) All experiments were performed in biological triplicate (n = 3). Symbols represent individual replicate values. Bars represent means. Error bars represent standard deviations. Samples were uninduced (-aTc, black) or induced (+aTc, white). Significance was determined by two-way ANOVA with Tukey's multiple comparisons test to determine adjusted p-values: (A) ns = non-significant, **** p <0.0001, (B) ns = non-significant, ** p = 0.0019, *** p = 0.0005, 0.0004, **** p <0.0001.
(TIFF)

**S6 Fig. c-di-GMP abundance and surface behaviors can be modulated in surface-associated cultures with inducible expression of phosphodiesterases and diguanylate cyclases.**
(A) Light production from a transcriptional reporter of the $scrABC$ surface-sensing operon ($P_{scrA}$-$luxCDABE$) in planktonic (black) and surface-associated (white) RIMD strains.
(B) Relative c-di-GMP abundance across the indicated strains grown either in liquid or on a solid surface. (C) Fold-change in c-di-GMP abundance in surface-associated $\Delta opaR$ RIMD carrying either an empty vector control (EV), an inducible $scrABC$ operon ($P_{tet}$-$scrABC$) in which ScrC functions as a phosphodiesterase, or an inducible allele of $scrC$ ($P_{tet}$-$scrC^{E554A}$) in which ScrC functions as a diguanylate cyclase. (D) Light production from a transcriptional reporter of the lateral flagellin promoter ($P_{lafA}$-$luxCDABE$) in surface-associated $\Delta scrABC$ RIMD carrying either EV or $P_{tet}$-$scrABC$. (E) Light production from $P_{lafA}$-$luxCDABE$ in surface-associated $\Delta opaR$ RIMD carrying either EV or $P_{tet}$-$scrC^{E554A}$. (F) Fold-change in c-di-GMP abundance in surface-associated $\Delta opaR$ RIMD carrying either EV, an inducible $P_{tet}$-$tpdA$ construct encoding the TpdA phosphodiesterase, or an inducible $P_{tet}$-$gefA$ construct encoding the GefA diguanylate cyclase. (G) Light production from $P_{lafA}$-$luxCDABE$ in surface-associated $\Delta scrABC$ RIMD carrying either EV or $P_{tet}$-$tpdA$. (H) Light production from $P_{lafA}$-$luxCDABE$ in surface-associated $\Delta opaR$ RIMD carrying either EV or $P_{tet}$-$gefA$. (A-H) All experiments were performed in biological triplicate (n = 3). Symbols represent individual replicate values. Bars represent means. Error bars represent standard deviations. (C-H) Black and white bars are uninduced (-aTc) and induced (+aTc), respectively. (A,D,E,G,H) RLU are bioluminescence normalized to constitutive mScarlet-I signal. (B,C,F) RFU is c-di-GMP controlled TurboRFP fluorescence normalized to constitutive AmCyan fluorescence. (C,F) aTc-induced values (white) are represented as the fold-change versus their corresponding uninduced values (black). (A,C-H) Significance was determined by two-way ANOVA with Sidak's multiple comparisons test to determine adjusted p-values: (A) ns = non-significant, **** p <0.0001, (C) ns = non-significant, * p = 0.0232, **** p <0.0001, (D) ns = non-significant, **** p <0.0001, (E) ns = non-significant, ** p = 0.0095, (F) ns = non-significant, * p = 0.0168, **** p <0.0001, (G) ns = non-significant, * p = 0.0395, (H) ns = non-significant, ** p = 0.0021. (B) Significance was determined by two-way ANOVA with Tukey's multiple comparisons test to

determine adjusted p-values: * p = 0.0273, **** p <0.0001.
(TIFF)

## Acknowledgments

We are grateful to all members of the Bassler Laboratory for insightful discussion, and we thank Ned Wingreen for critical advice regarding quantitative aspects of the assays developed here.

## Author Contributions

**Conceptualization:** Francis J. Santoriello, Bonnie L. Bassler.

**Data curation:** Francis J. Santoriello, Bonnie L. Bassler.

**Formal analysis:** Francis J. Santoriello, Bonnie L. Bassler.

**Funding acquisition:** Bonnie L. Bassler.

**Investigation:** Francis J. Santoriello, Bonnie L. Bassler.

**Methodology:** Francis J. Santoriello.

**Project administration:** Bonnie L. Bassler.

**Resources:** Francis J. Santoriello, Bonnie L. Bassler.

**Supervision:** Bonnie L. Bassler.

**Validation:** Francis J. Santoriello, Bonnie L. Bassler.

**Visualization:** Francis J. Santoriello, Bonnie L. Bassler.

**Writing – original draft:** Francis J. Santoriello, Bonnie L. Bassler.

**Writing – review & editing:** Francis J. Santoriello, Bonnie L. Bassler.

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
