## [Decision Letter · Decision Letter 0]

2 May 2024

Dear Dr Bassler,

Thank you very much for submitting your Research Article entitled 'The LuxO-OpaR quorum-sensing cascade differentially controls Vibriophage VP882 lysis-lysogeny decision making in liquid and on surfaces' to PLOS Genetics.

The manuscript was fully evaluated at the editorial level and by independent peer reviewers. The reviewers appreciated the attention to an important problem, but raised some substantial concerns about the current manuscript. Based on the reviews, we will not be able to accept this version of the manuscript, but we would be willing to review a much-revised version. We cannot, of course, promise publication at that time.

If you decide to revise the manuscript for further consideration at PLOS Genetics, please aim to resubmit within the next 60 days, unless it will take extra time to address the concerns of the reviewers, in which case we would appreciate an expected resubmission date by email to plosgenetics@plos.org.

We are sorry that we cannot be more positive about your manuscript at this stage. Please do not hesitate to contact us if you have any concerns or questions.

Yours sincerely,

Ankur B. Dalia

Academic Editor

PLOS Genetics

Lotte Søgaard-Andersen

Section Editor

PLOS Genetics

As you will see below, all three reviewers were enthusiastic about this submission and believe it is a highly valuable contribution to the field. But they also highlight a number of issues that need to be addressed. In particular, the reviews highlight concerns with the newly developed Q-induction assay used to assess lysogeny, and have provided actionable suggestions on how to address these concerns and further validate the use of this assay with another approach.

Reviewer's Responses to Questions

**Comments to the Authors:**

Reviewer #1: Major comments

1. Despite the argument presented for why the large differences in phage adsorption efficiency between strains don’t impact the results, at a minimum, it clouds interpretation of the data. Being only a 10-minute phage adsorption assay (Figure S3), one must suspect significant differences in phage receptor expression between strains. However, nowhere in the manuscript is the receptor even mentioned. What is the receptor, and does its expression vary between the different strains tested? Is receptor expression cell density dependent, or planktonic versus surface dependent? Why wasn’t phage adsorption measured for surface-grown strains?

2. The Discussion section states several times that the phage can adsorb to cells grown at any density, but how can this be, given that it does not absorb well to WT and luxOD61E in Figure S3? Perhaps you mean that it can adsorb to at least a fraction of the cells regardless of culture density.

3. The Q-induction assay for lysogeny of surface-grown bacteria needs to be validated in some way, as was done for planktonic cells. Otherwise, how can we trust the lysogeny data presented?

Minor comments

Lines 125, 211, 218, etc: Your use of “surprisingly” is not scientific and distracts the reader. I would delete all such uses.

Line 171: This is more likely due to a polar effect of the antibiotic resistance cassette. Either way, such speculation isn’t necessary. Replace with “due to unknown reasons.”

Line 367: Missing a hyphen.

Reviewer #2: The authors present a thorough and deep assessment of the role of quorum sensing in VP882 phage lysis-lysogeny decision. Through very detailed and thorough genetic experiments, the authors demonstrate two key findings: 1) the QS circuit components OpaR and the Qrrs regulate lysogeny at HCD and LCD, respectively, thus only at LCD will lysogeny occur, and 2) the surface-sensing system of ScrABC blocks lysogeny on surfaces at HCD, also controlled by OpaR. I am impressed with the fine-level detail of the genetic dissection of the pathways, as well as the assays that validate the QS states, measurements of lysogeny, expression of Qrr2, and measurements of cdGMP levels. I have only a few comments for clarity and an experimental suggestion.

Major comments:

• Line 173: I wondered when reading: why did the authors not measure lysogeny vs lysis by qPCR using the formation/level of attL or attR vs attP. I understand that the authors developed the Pbad-q assay to measure lysis to calculate lysogenic conversion, and that works here. But it may be useful (either in this manuscript or in the future) to use the above-mentioned method for direct comparison of lysis and lysogeny both by qPCR (which can then also measure copy number during lytic replication).

• Line 206: When infecting RIMD strains, what OD were they at in the beginning of the experiment? Methods say 1:100 dilution of an overnight culture. I suspect this isn’t actually “LCD” because the WT behaved like the HCD strain, and thus that result implies that the density of the infected cultures was similar to the HCD state. This should be clarified in the results. Would the result be different if the overnight cultures were diluted 1:10,000? (Line 521 methods). This seems likely as this is what appears to happen in subsequent experiments (lines 252, 282).

• Line 333: This should be a new results subsection connecting all the cdGMP results.

Minor comments:

• Lines 263-269 seem repetitive with the info in the introduction.

• It would help with reading/clarity if the graphs throughout were labeled with “planktonic” or “surface-associated”.

Reviewer #3: Review of the manuscript “The LuxO-OpaR quorum-sensing cascade differentially controls Vibriophage VP882 lysis-lysogeny decision making in liquid and on surfaces” by Santoriello and Bassler

Summary of the manuscript

This work follows on previous works of the Bassler lab which have shown that phage phiVP882 can switch from lysogeny to lysis through sensing of a quorum-sensing signal (DPO) produced by the host. This occurs through activation of a phage encoded receptor (phiVqmR) homologous to the host receptor which senses the signal and leads to the production of an antirepressor. In a follow-up work they have also shown that this induction is enhanced by a factor regulated by other QS systems of the host (LuxO).

In this work, Santoriello and Bassler set out to find host factors which affect the lysis-lysogeny decision of the phage upon infection. Lacking the ability to perform PFU and CFU measurements, they have developed alternative assays to assess the number of lytic phages and the frequency of lysogens in a population. They found that LuxO activity, regulated by the triple converging QS systems characterizing many vibrios (and studied by the Bassler lab), is a strong modulator of the level of lysogeny, with lysogeny enhanced by conditions of low cell density. They find that this effect is differentially regulated in liquid medium and in surface association. In liquid medium, this effect requires only LuxO~P dependent activation of the small qrr RNAs, while in surface association LuxO~P transduces the signal to OpaR which controls c-di-GMP levels through the Scr system, and lysogeny is induced at low c-di-GMP level.

General assessment

This is a very intriguing work, and is definitely worth publishing in PLOS Genetics, once all experimental issues are resolved. However, there are several major issues with the lysogeny assay used by the authors which will require further validation of the results. There are also several issues with the interpretation of the results and their presentation.

Major issues - Lysogeny assay

To assess lysogenization rate (or propensity), the authors perform the following assay (as detailed in the methods section). They dilute ON grown bacteria 1:100 and infect them at a MOI of 0.1. They then let bacteria and phages grow for 24 hours.

All mutations where done in a strain encoding the pro-lytic q gene on an IPTG inducible promoter. After the 24 hours of growth, cells are diluted 1:100 into media where phages cannot infect and grown further to mid-log, prior to induction of the host-encoded q gene by IPTG and assessment of lysogeny.

There are several issues with this assay as I discuss below:

1. Does this assay actually measure lysogenization frequency at all? Phages and bacteria are left for 24 hours after infection, during this time there is clearly a lot of infections occurring resulting in lysogeny, but also in a lot of lysis and re-infection. The assay therefore is a cumulative measurement of lysogeny during the whole infection process. The final number (and maybe also frequency) of lysogens in such an assay is optimized by mutants which are not fully lysogenic. In fact, a mutant which leads to full lysogeny and infected at MOI of 0.1 would necessarily result in 10% frequency of lysogens and not in 100%, as all infections would immediately lead to lysogeny. Therefore, in this case, the strain with lower lysogenization frequency would achieve a higher score, showing that this assay may be flawed in principle.

It is important to try and assay the impact of the mutants on lysogeny in a single round of infection. I suggest allowing the cells and phages to interact for a limited time (say 30 minutes) and not for 24 hours to assess the level of lysogenization there. Now, I am aware of the fact that the frequency of resulting lysogens may be too low to appropriately measure with the q induction method, but it is worth trying. Additionaly, I suggest adding a different, complementary, assay in any case, as detailed below.

2. Why are there non-lysogens at all? The measurement of lytic particles suggests that viral particles reach very high numbers by 24 hours of infection. If this is the case, it is not clear at all to me why non-infected cells remain, as we would expect only lysogens to be resistant to infection. I could see several options. First, the non-infected cells may be mutants that arose during the 24 hours. Second, the community might go into a non-infective state at a certain timepoint (e.g, upon entering stationary) maintaining cell that are uninfected, though sensitive. Further, in some mutants 100% of bacteria became lysogens, indicating that all cells have been infected. If there is no major difference in the ability of phages to infect the different mutants (as claimed in the manuscript), how is it that in certain mutants there remain uninfected cells but not in others? Specifically, the mutants which most increase lysogenization frequency are those that show 100% infection, which adds to the mystery.

It is important to distinguish between the above options. With the current assay of 24 hours of infection, it may be that a major effect of the different host mutations is the time when the cells enter the non-infected state, or the tendency of resistance mutants to arise in different host background and not the tendency to lysogenize.

Also, in this respect, it would be good to provide some understanding of the total number of cells that survive after 24 hours of infection and not only the frequency of infected and non-infected cells. A supp figure of a plate reader graph showing infections of WT, high cell density, and low cell density mutants in liquid would also help understand the difference between the strains.

Again, all of these issues stress the importance of monitoring a single infection round and not the result of a culminative infection process after 24 hours.

3. How sensitive is the q-induction assay? The authors perform the appropriate calibration of the assay to a set of cocultures of lysogens and non-lysogens at different frequencies. However, given the limits of the system (low reliability below ~5-10%), I think it would be informative to compare these results (at least of several selected mutants, with a clear difference in lysogenization frequency) to alternative quantification methods. For example, the authors can perform a qPCR (of the same gene they use for assaying lytic particles) on the genomic extraction of a population. I would expect to see clear correlation between the frequency of lysogens in the two assays. The qPCR assay may be more reliable at low frequencies and may therefore allow measuring lysogeny after a single round of infection.

Minor issues

========

1. Given the importance of the Vqm system to induction, it is surprising that the authors haven’t checked the impact of the host Vqm system on lysis-lysogeny. Similarly, does addition of DPO affect the decision? This is not necessary for the manuscript, but would improve it.

2. It is also surprising that the wild-type is non-lysogenic. My guess is that a 100-fold dilution from overnight never allows the cells to really go into the LCD state? It might be interesting (but not necessary) to dilute much more (say 10,000) and let the cells grow for some time to reach the same density

---

## [Editor Report · Decision Letter 1]

12 Jul 2024

Dear Dr Bassler,

We are pleased to inform you that your manuscript entitled "The LuxO-OpaR quorum-sensing cascade differentially controls Vibriophage VP882 lysis-lysogeny decision making in liquid and on surfaces" has been editorially accepted for publication in PLOS Genetics. Congratulations!

Yours sincerely,

Ankur B. Dalia

Academic Editor

PLOS Genetics

Lotte Søgaard-Andersen

Section Editor

PLOS Genetics

Comments from the reviewers (if applicable):

**Data Deposition**

http://datadryad.org/submit?journalID=pgenetics&manu=PGENETICS-D-24-00342R1

**Press Queries**

---

## [Editor Report · Acceptance letter]

25 Jul 2024

PGENETICS-D-24-00342R1 

The LuxO-OpaR quorum-sensing cascade differentially controls Vibriophage VP882 lysis-lysogeny decision making in liquid and on surfaces 

Dear Dr Bassler, 

We are pleased to inform you that your manuscript entitled "The LuxO-OpaR quorum-sensing cascade differentially controls Vibriophage VP882 lysis-lysogeny decision making in liquid and on surfaces" has been formally accepted for publication in PLOS Genetics! Your manuscript is now with our production department and you will be notified of the publication date in due course.

With kind regards,

Lilla Horvath

PLOS Genetics

On behalf of:
